# Correlation measurement of propagating microwave photons at millikelvin

**Aarne Keränen** [1,3], **Qi-Ming Chen** [1,3] ✉, **András Gunyhó** [1], **Priyank Singh**[1], **Jian Ma**[1], **Visa Vesterinen** [2], **Joonas Govenius** [2] & **Mikko Möttönen** [1,2] ✉

Microwave photons are essential carriers of quantum information in several promising platforms for quantum computing. However, measurement of the quantum statistical properties of microwave photons is demanding owing to their low energy relative to thermal fluctuations of any room-temperature detector, and phase-insensitive voltage amplification necessarily adds noise. Here, we overcome this trade-off with a nanobolometer that directly measures the photon statistics at millikelvin. Using a cryogenic temperature-controlled blackbody radiator, we demonstrate the detection of the mean photon number $\langle \hat{n} \rangle$ and reveal the expected photon number variance $(\Delta n)^2 = \langle \hat{n} \rangle (\langle \hat{n} \rangle + 1)$, following the Bose–Einstein distribution. By engineering the coherent and incoherent proportions of the input field, we observe a transition between super-Poissonian and Poissonian statistics from the bolometric second-order correlation measurements. This technique is poised to serve in fundamental tests of quantum mechanics and function as a scalable readout solution for a quantum information processor.

Microwaves with the wavelength ranging from several millimeters to several meters are extensively used for communication, sensing, and power applications[1]. In the realm of quantum physics, many competitive platforms for quantum computation, such as superconducting quantum circuits[2], quantum dots[3], and circular Rydberg atoms[4], are also operated in this regime. These devices have to be cooled down to a cryogenic temperature where the quantum statistical properties of microwave photons become prominent over thermal noise. Weak microwave signals are subsequently amplified from the cryogenic to the room temperature stages for detection, which normally adds more than 10 noise photons as referred to the input of a cryogenic high-electron-mobility transistor (HEMT) amplifier[5]. For many years, attempts have been made to circumvent the amplification noise by a variety of parametric processes[6–15]. An ideal degenerate parametric amplifier may be noiseless at a price of squeezing the quantum states in one quadrature. Breaking the degeneracy solves the phase-nonpreserving problem, but it must add at least half of a quantum of noise to each quadrature of the input signal as required by quantum

mechanics[16]. The quantum statistical properties of microwave photons can only be inferred from hours-long heterodyne detection of the noise itself and the amplified noisy signal, and through a complicating deconvolution procedure[17].

An alternative but yet to-be-demonstrated approach searches for a cryogenic detector that carries out the projective quantum measurements directly at the millikelvin temperature without amplification noise. In addition to the early attempts of bistable detectors[18], to date the most successful protocol uses the two states of a qubit as a pointer[19,20]. With a certain repetition rate, the qubit can be operated as a click detector that gets excited when absorbing an incoming photon within a narrow frequency range. Recently, a superconductor-normal metal-superconductor (SNS) junction-based nanobolometer has been invented as a broadband cryogenic microwave power detector with record-breaking speed and sensitivity[21–24]. This nanobolometer requires no pulse sequences for operation and has the potential to be used as a photon-number resolved microwave photon detector. Here, we report a measurement technique with the nanobolometer to

[1]QCD Labs, QTF Centre of Excellence, Department of Applied Physics, Aalto University, Aalto, Finland. [2]VTT Technical Research Centre of Finland Ltd. & QTF Centre of Excellence, P.O. Box 1000, Espoo, Finland. [3]These authors contributed equally: Aarne Keränen, Qi-Ming Chen. ✉e-mail: qiming.chen@aalto.fi; mikko.mottonen@aalto.fi

measure the quantum statistical properties of propagating microwave photons at the cryogenic temperature.

The nanobolometer, as shown in Fig. 1a, has a transmission-line filter that selects the bandwidth of interest. The filter is terminated by a normal-metal nanowire with an ~50 $\Omega$ resistance, i.e., the absorber. The absorber transforms the electromagnetic energy of the incident photons to a significant increase of the electron temperature in the nanowire. This nanowire extends through several superconducting islands and forms an array of short SNS junctions that are embedded in a sub-gigahertz probe resonator, i.e., the thermometer[25] (see Supplementary Fig. 1). The hot electrons weaken the coherent Andreev reflections at the SN and NS interfaces of each junction, and cause a downshift of the mean resonance frequency of the thermometer. The amount of the shift is determined by the mean absorbed power, which for a given frequency and bandwidth can be characterized by the mean photon number of the input field, $\langle \hat{n} \rangle$, considered in the unit of per second per hertz as it is the conventional unit for describing a stationary beam of electromagnetic field (see Supplementary Note 1). The broadening of the lineshape is related to the photon number variance, $(\Delta n)^2 = \langle \hat{n}^2 \rangle - \langle \hat{n} \rangle^2$, similar to the Doppler broadening in laser spectroscopy[26] (see "Methods"). The nanobolometer therefore allows direct readout of the first two orders of photon number moments, $\langle \hat{n} \rangle$ and $\langle \hat{n}^2 \rangle$, from the averaged scattering response of the thermometer without any amplification of the input field. Note that the incident and probe fields are not directly electrically coupled to each other, owing to the ground port between the absorber and the thermometer, and have several gigahertz difference in frequency.

Here, we use a cryogenic blackbody radiator to generate thermal microwave photons for bolometric measurement. By controlling the local temperature of the radiator, $T$, we achieve an in-situ control of the mean radiation photon number according to the Planck's law,

$\langle \hat{n} \rangle = 1 / \{ \exp[h f_h / (k_B T)] - 1 \}$, where $f_h$ is the input photon frequency, and $h$ and $k_B$ are the Planck and Boltzmann constants, respectively. With the decrease of $\langle \hat{n} \rangle$, the photon number variance converges from a quadratic function, $(\Delta n)^2 \approx \langle \hat{n} \rangle^2$ at $\langle \hat{n} \rangle \gg 1$, to a linear relation, $(\Delta n)^2 \approx \langle \hat{n} \rangle$ at $\langle \hat{n} \rangle \ll 1$[27]. Correspondingly, the photon statistics crosses over the boundary between the classical and the quantum regimes of the radiation field and converges to the shot-noise limit, indicating the discreteness of photons. This process is precisely described by the Bose-Einstein distribution of indistinguishable bosons, where $(\Delta n)^2 = \langle \hat{n} \rangle (\langle \hat{n} \rangle + 1)$.

We also use a weak coherent input field as the reference of the bolometric measurement, where $(\Delta n)^2 = \langle \hat{n} \rangle$ by definition. In addition to the different scaling laws of $(\Delta n)^2$, a stark contrast between the thermal and coherent photons lies in the second-order correlation function, $g^{(2)}(\tau)$, with $\tau$ being the time delay. It characterizes the temporal separation between two successive photons in a propagating field. At zero delay, we have $g^{(2)}(0) = 1 + [(\Delta n)^2 - \langle \hat{n} \rangle] / \langle \hat{n} \rangle^2$. A perfect coherent field obeys Poissonian statistics with $g^{(2)}(0) = 1$, while thermal photons are super-Poissonian with $g^{(2)}(0) = 2$. Considering that $g^{(2)}(\tau) = 1$ for $\tau \to \infty$, an observation of $g^{(2)}(0) > 1$ indicates the photon bunching effect where the photons tend to propagate in bundles.

## Results

### Characterization of the nanobolometer

In our experiment, we first sweep the probe frequency from $f_p = 510$ to 530 MHz to identify the resonance frequency of the thermometer (Fig. 1b). At the minimum probe power, $P_p = -140$ dBm defined at the thermometer input by assuming an 80 dB attenuation from the source (see Supplementary Fig. 2), we extract the resonance frequency as $f_r = 524$ MHz, as well as the external and the total energy decay rates $\gamma_c = 4.8 \, \mu s^{-1}$ and $\gamma = 18.7 \, \mu s^{-1}$, respectively, using the circle-fit method[28].

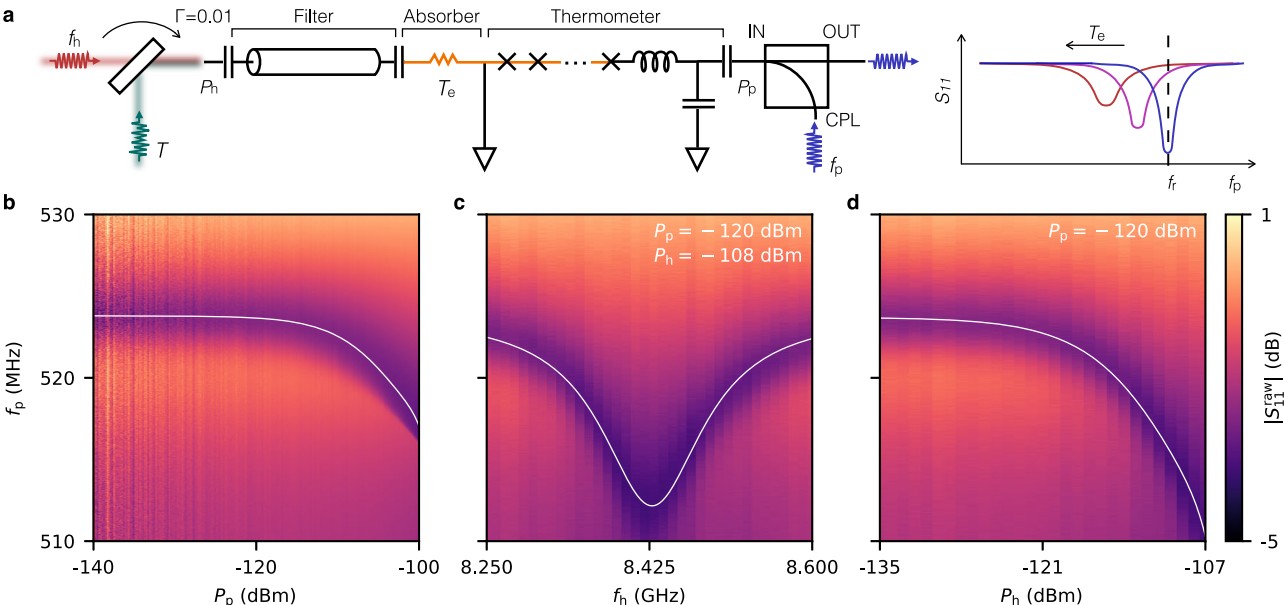

**Fig. 1 | Measurement scheme and characterization of the nanobolometer with coherent input. a** Schematic diagram of the experimental setup. A beam splitter with a transmission rate $\Gamma = 0.01$ combines a coherent microwave field at frequency $f_h$ with a thermal field characterized by the radiation temperature $T$. The transmitted field is filtered and subsequently absorbed by a resistive absorber, causing a change of the electron temperature, $T_e$, in the normal-metal nanowire. It leads to a decrease of the critical current of the SNS junctions (crosses) and hence a downshift in the resonance frequency of the thermometer. This shift is measured by a weak probe field at frequency $f_p$, which is coupled to the thermometer via a directional coupler. **b** Reflection magnitude of the thermometer as a function of the probe power at the sample, $P_p$, and the probe frequency, $f_p$. The solid curve represents a cubic fit of the resonance frequency. **c** Reflection magnitude of the thermometer as a function of the input frequency, $f_h$, and the probe frequency. The probe and input powers are fixed at $P_p = -120$ dBm and $P_h = -108$ dBm, respectively. The solid curve is a Lorentzian fit of the resonance frequency, which indicates a passband of FWHM = 133 MHz around the central frequency $f_0 = 8.428$ GHz. **d** Reflection magnitude of the thermometer as a function of the input power at the sample, $P_h$, and the probe frequency, $f_p$. Here, the probe power is fixed at $P_p = -120$ dBm and $f_h$ is set to the center frequency of the filter. The solid curve is a cubic fit of the resonance frequency. The data is normalized to 0 dB at 530 MHz probe frequency in (**b**)–(**d**).

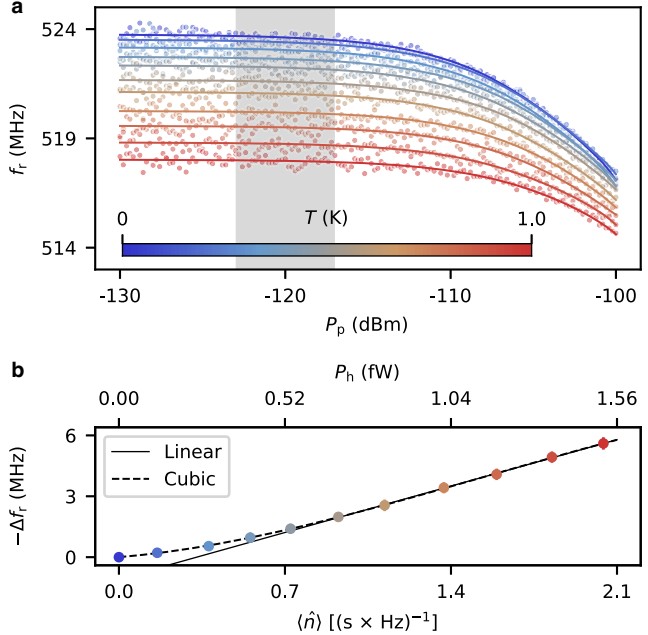

**Fig. 2 | Photon-number resolved frequency shift. a** Resonance frequency of the thermometer, $f_r$, as a function of the probe power, $P_p$, at different indicated temperatures of the blackbody radiation, $T$. The dots and solid curves are the corresponding experimental data and cubic fits, respectively. **b** Frequency shift, $\Delta f_r$, in the region highlighted with gray in (**a**) as a function of the mean photon number (power) of the thermal radiation field, $\langle \hat{n} \rangle$ ($P_h$). The dots and error bars represent the mean values and standard deviations of the raw data in the selected range. The dashed black line shows a cubic fit to the full data and the solid black line represents a linear fit to the data where $\langle \hat{n} \rangle \geq 1$. The top horizontal axis indicates the radiation power in the entire 133 MHz bandwidth of the input filter.

We observe that $f_r$ remains almost invariant at $P_p < -110$ dBm, but starts to bend downwards at higher probe power together with the lineshape changing from a symmetric into an asymmetric form. Possible sources of this nonlinearity are the coherent and incoherent Andreev reflections in the short SNS junctions, leading to the Josephson effect[29] and electrothermal feedback[22,30], respectively.

We then set the probe power at $P_p = -120$ dBm, and characterize the filter passband with a coherent input signal. Here, we sweep the input frequency from $f_h = 8.25$ to $8.60$ GHz with a fixed power $P_h = -108$ dBm defined at the filter input by fitting an 88 dB attenuation from the source (Fig. 1c). Depending on the frequency of the input signal as compared to the passband, the resonance frequency of the thermometer, $f_r$, shifts as a Lorentzian function of $f_h$. The center frequency and bandwidth of the filter are determined to be $f_0 = 8.428$ GHz and FWHM = 133 MHz, respectively.

Subsequently, we fix the input frequency at the center of the passband, $f_h = f_0$, and vary its power from $P_h = -135$ to $-107$ dBm (Fig. 1d). The other parameters are kept identical to those in Fig. 1c. We observe that $f_r$ decreases with increasing $P_h$, demonstrating the functionality of the nanobolometer as a power detector. In contrast to Fig. 1b, the lineshape remains symmetric owing to a different mechanism than the nonlinearity. Here, the shift is caused by the increase of the electron temperature, $T_e$, in the short SNS junctions, leading to an increase of the inductance in the resonant circuit. The relative shift, $\Delta f_r$, compared with the highest value of $f_r$ can be well fitted by a cubic function of $P_h$. Here, the input photon number, $\langle \hat{n} \rangle = P_h/(\mathrm{FWHM}hf_0)$, is estimated to be $0$–$27$ photons/(s × Hz).

## Photon number resolution
For a more precise benchmark of the frequency shift with different $\langle \hat{n} \rangle$, we use a blackbody radiator as the nanobolometer input. We sweep

the radiation temperature, $T$, from the base temperature to ~1 K while keeping the mixing chamber (MXC) temperature below 60 mK. The corresponding input photon number, $\langle \hat{n} \rangle$, can be precisely determined by the Planck's law, which varies from 0 to 2 photons/(s × Hz). At each radiation temperature, we sweep both $P_p$ and $f_p$ and identify the resonance frequency, $f_r$, at the dip of the averaged reflection magnitude. Here, the intermediate-frequency (IF) bandwidth is set to 200 Hz with 10 averages. Figure 2a summarizes the measurement results. A clear decrease of $f_r$ with increasing $T$ is observed, which is consistent with the observation with a coherent input field as shown in Fig. 1d.

Averaging the relative frequency shift within an ±3 dB range around $P_p = -120$ dBm, we observe a cubic relation between $\Delta f_r$ and $\langle \hat{n} \rangle$ (Fig. 2b). At the minimum value of $\langle \hat{n} \rangle = 0.16$ photons/(s × Hz), we obtain the mean and standard deviation of $-\Delta f_r$ as 0.22 MHz and 0.19 MHz, respectively, corresponding to a coefficient of variation (CV) less than 1. We therefore attribute 0.16 photons/(s × Hz) as an upper bound of the photon number resolvedness of the nanobolometer. The value of CV decreases monotonically with $\langle \hat{n} \rangle$ and approaches 0.05 at the highest photon number. In fact, this resolution may be pushed to an even smaller value by decreasing the IF bandwidth or increasing the number of averages.

We note that the relatively small photon number resolution is benefited from the broad bandwidth of the blackbody radiation that covers the entire 133 MHz passband of the input filter. The corresponding input-field-independent power resolution of the nanobolometer is 119 aW, indicating a considerable amount of individual photon wavepackets per second, $2.1 \times 10^7$ photons/s. In this regard, the nanobolometer reported here is still far from resolving each individual photon emission event in the radiation that has been demonstrated in the optical regime[31]. Fortunately, the observed power resolution is adequate for resolving the quantum statistical properties of propagating microwave photons that are conventionally defined by the arriving photons per second per hertz[27].

## Photon number variance
The quantum statistical properties of thermal states are fully determined by the mean photon number, $\langle \hat{n} \rangle$, and its variance, $(\Delta n)^2$. With a weak radio-frequency (RF) probe field applied to the thermometer, these two values can be respectively obtained from the fitted resonance frequency, $\mu$, and the broadening of the spectral linewidth, $\sigma^2$ (see Supplementary Note 3). Here, we sweep the radiation temperature, $T$, from the base temperature to ~2 K and measure the averaged reflection coefficient, $S_{11}^{ave}$, at a fixed probe power, $P_p = -120$ dBm. We first fit the theoretical curve to the data at the minimum $T$ as a reference curve for zero broadening ($\sigma^2 = 0$). We then allow $\sigma^2$ to be a fitting parameter and extract its value at different $T$ by comparing the measurement results with this reference (see Supplementary Note 5). The photon number variance is obtained by assuming a linear relation $\Delta n = \alpha\sigma$, where $\alpha = 1.97$ photons/MHz is a fitted scaling factor.

At a sufficiently small photon number, where $\langle \hat{n} \rangle$ is determined by the Planck's law, the extracted photon number variance, $(\Delta n)^2$, is almost equal to $\langle \hat{n} \rangle$, as shown in the inset of Fig. 3a. It reveals the particle nature of photons in the form of shot noise, as described by the Poisson distribution. On the other hand, $(\Delta n)^2$ can be well approximated by a quadratic function, $\langle \hat{n} \rangle^2$, at large $\langle \hat{n} \rangle$. These two limits are connected by the Bose-Einstein distribution, which predicts $(\Delta n)^2 \approx \langle \hat{n} \rangle (\langle \hat{n} \rangle + 1)$ for indistinguishable bosons in thermal states. Our experimental results in Fig. 3a show an excellent agreement with the theoretical expectation. It reveals the transition between the quantum and classical regimes of a microwave field with the increase of the photon number.

For comparison, we replace the thermal input field by a coherent signal at $f_0$. We sweep the input power from $P_h = -128$ to $-108$ dBm, and extract $\langle \hat{n} \rangle$ and $(\Delta n)^2$ with almost identical fitting parameters as for the thermal fields (see Supplementary Fig. 3). We observe a qualitatively

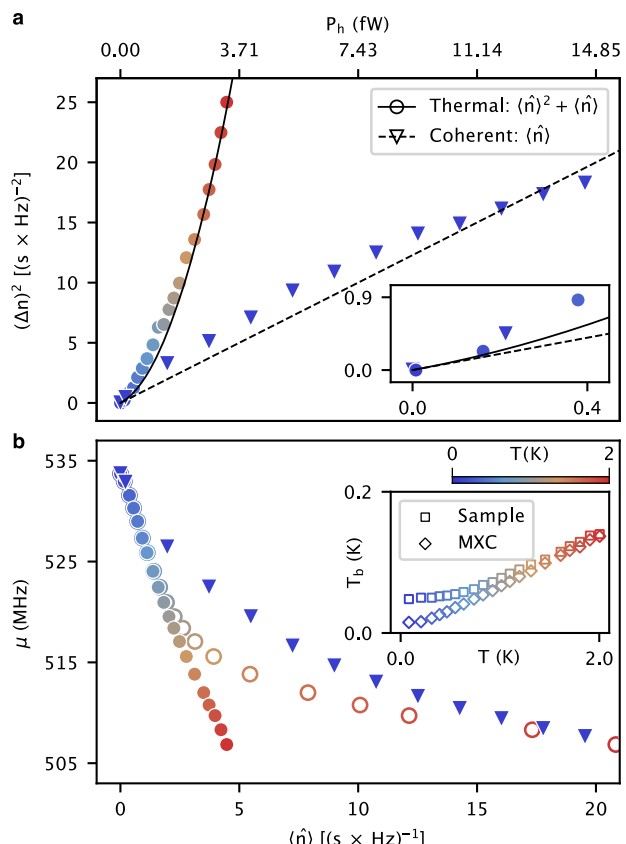

**Fig. 3 | Photon number variance extracted from the reflection spectrum.**
**a** Photon number variance, $(\Delta n)^2$, as a function of the mean input photon number (power), $\langle \hat{n} \rangle$ ($P_h$), for the thermal (circles) and coherent (triangles) input fields. The solid and dashed lines represent the theoretical expectation for thermal and coherent states, respectively. The thermal photon number is obtained by Planck's law, and the coherent photon number is obtained by assuming an 88 dB attenuation in the input line as a scaling parameter. The detection efficiency of the nanobolometer is assumed to be unity to minimize the number of fitting parameters. The inset shows the region near the origin. **b** Extracted resonance frequency, $\mu$, as a function of $\langle \hat{n} \rangle$ for thermal (circles) and coherent (triangles) input fields. The open circles are corrected results by considering the phonon temperature change, $T_b$, during the temperature sweep. The inset shows the records of two different sensors that are mounted at the sample holder and the MXC, respectively. The former is lower-bounded by ~45 mK due to self-heating. The top horizontal axis indicates the radiation power in the entire 133 MHz bandwidth of the input filter.

different statistics of photons between the thermal and coherent input fields, as shown in Fig. 3a. Here, $(\Delta n)^2$ remains as a pure linear function of $\langle \hat{n} \rangle$ in the entire range of the power sweep, corresponding to a mean photon number from 0 to 19. This observation is in agreement with the Poisson statistics of the input photons in a coherent state. The excellent agreement between theory and experiments for both thermal and coherent states with almost the same fitting parameter demonstrate the correctness of our measurement result.

Figure 3 b shows the extracted value of $\mu$ for both thermal and coherent inputs. The data are in quantitative agreement with each other for $\langle \hat{n} \rangle < 2$, indicating an almost linear relation between the frequency shift of the thermometer and the input power as observed in Fig. 2b. However, a noticeable discrepancy is observed at the higher input power. Since the temperature of the blackbody radiator, $T$, is significantly higher than the MXC temperature in this range, we attribute the observed discrepancy to the heating of the phonon bath temperature, $T_b$ (see Supplementary Note 2). The latter is recorded in the inset of Fig. 3b. We then consider a 6th-order polynomial correction of the net heating power due to the phonon temperature change, i.e.,

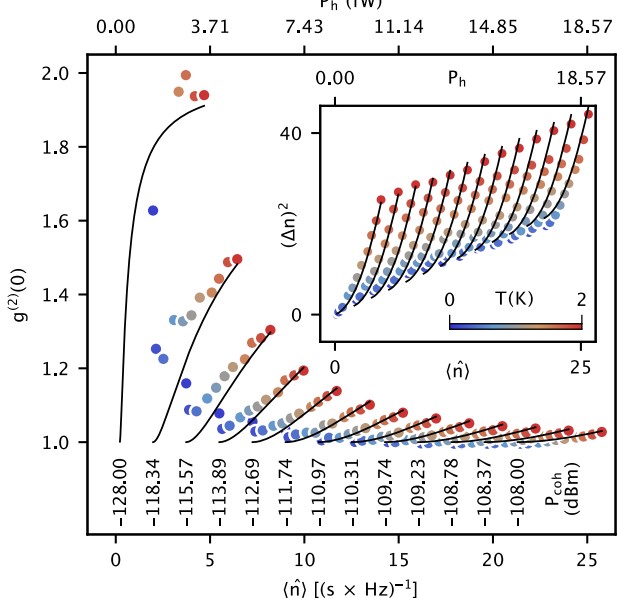

**Fig. 4 | Second-order correlation for different photon states.** The proportions of the coherent and thermal parts in the input field are controlled by varying the power of the coherent field, $P_{coh}$, and the radiation temperature, $T$. Here, the solid curves correspond to the theoretical expectations at fixed values of $P_{coh}$ but with a varying $T$. $P_h$ denotes the total input power. The inset shows the raw data of the photon number and variance that are used to calculate $g^{(2)}(0)$. The top horizontal axis indicates the radiation power in the entire 133 MHz bandwidth of the input filter.

$P = \beta T_b^6 + P_h$, for ultra-thin films[32]. A qualitative agreement of $\mu$ for the two types of photons is achieved with $\beta = 1.84$ nW/K⁶, as shown in Fig. 3b. A quantitative agreement can be achieved by assuming an ~50% absorption rate of the nanobolometer, which is not taken into account throughout this study.

## Photon correlation

The shown ability of resolving the mean photon number, $\langle \hat{n} \rangle$, and its variance, $(\Delta n)^2$, indicates a convenient way of measuring the zero-delay second-order correlation functions of the incident microwave field, $g^{(2)}(0)$. In our experiment, we control the coherent and incoherent proportions of the input field by adjusting their powers at the beam splitter inputs (Fig. 1a), and repeat the photon variance measurement as shown in Fig. 3.

Figure 4 summarizes the correlation measurement result as well as the raw data of $\langle \hat{n} \rangle$ and $(\Delta n)^2$. At a fixed power of the coherent field, $P_{coh}$, the correlation function, $g^{(2)}(0)$, increases with the radiation temperature, $T$, from 1 to a value slightly below 2. It reveals the loss of coherence as the thermal proportion of the field increases. On the other hand, $g^{(2)}(0)$ converges to 1 with increasing $P_{coh}$. This is consistent with our expectation since an ideal coherent field leads to $g^{(2)}(0) = 1$. The measured value of $g^{(2)}(0)$ is larger than the theoretical expectation by a few percent where the coherent photon number is below 1 photons/(s × Hz). We attribute this loss of coherence to the 88 dB attenuation in the input line, where the blackbody radiation at different thermalization stages necessarily impinges on the input signal (see Supplementary Note 4). Overall, the observed $g^{(2)}(0)$ shows a quantitative agreement with the theoretical expectation. It is directly obtained from the resonance frequency and linewidth of the averaged scattering response without amplification and the corresponding noise. This is the key advantage of using cryogenic detectors for microwave photon correlation measurement compared with the combination of quantum limited amplifiers and room-temperature linear detectors[33,34].

## Discussion

The bolometric measurement of the second-order photon correlation function demonstrates the ability of using cryogenic detectors to characterize the quantum statistical properties of microwave radiation. This method implements projective quantum measurements directly at the millikelvin stage, and therefore only requires a data bus from the cryogenic to the room temperature stages to transfer the classical measurement outcome. Multiple nanobolometers can be multiplexed with the same bus[35]. The combination of long and short SNS junctions allows the RF operation of the nanobolometer, which plays a key role in reading out the photon number variance. This configuration also pushes the bandwidth of cryogenic detectors to the convenient sub-10 GHz regime as compared to other designs such as transition edge sensor (TES)[36], kinetic inductance detector (KID)[37], and superconducting nanowire single photon detector (SNSPD)[38]. By combining this technique with correlation spectroscopy[26], it is possible to measure even higher-order correlation functions of microwave photons with a single nanobolometer. An alternative upgrade for higher-order correlation measurements is to combine the nanobolometers with microwave beam splitters[39] in a similar way of the well established optical experiment setup.

In the future, the capability of microwave correlation measurements may advance the current understandings of the cosmic microwave background (CMB)[40] and resolve a possible distortion of the CMB frequency spectrum from a pure blackbody radiator. In addition, the small physical footprint and ultralow power consumption of the nanobolometer[23,24] make it suitable for millikelvin integration. Integrating high-performance microwave photon generators[41] and detectors[42] at millikelvin enables an isolated cryogenic environment for quantum information processors. It opens up new possibilities of controlling and reading out the quantum information carried by microwave photons with less noise, latency, and power consumption.

## Methods

### Sample preparation

The nanobolometer used in this study consists of an 7.5 mm-long Nb coplanar waveguide filter, an 150 nm-wide 1 μm-long $AuPd_x$ absorber, and an 7-island Al-$AuPd_x$-Al junction thermometer[22,23]. The superconducting islands are 300 nm wide and 2.4 μm long, and they are evenly spaced by a 300 nm gap. The designed thicknesses of the Si substrate and the $SiO_2$, Nb, $AuPd_x$, and Al layers are 675 μm, 300 nm, 200 nm, 30 nm, and 100 nm, respectively (see Supplementary Note 3 for circuit description and Supplementary Note 4 for experimental setup).

### Averaged response

The instantaneous reflection coefficient of the thermometer is described as $S_{11} = 1 - e^{i\phi}\gamma_c / [(\gamma/2) + i\Delta]$ [43], where $\gamma_c$ and $\gamma$ are the external and the total energy decay rates, and $\Delta = 2\pi(f_r - f_p)$ is the detuning between the resonance and the probe frequencies. The parameter $\phi$ describes the asymmetry of the resonance. We consider a Gaussian distribution of the resonance frequency, $f_r \sim \mathcal{N}(\mu, \sigma^2)$, which is caused by the statistical properties of the input photons (see Supplementary Notes 1 and 2). The averaged thermometer response is thus given by

$$S_{11}^{ave} = 1 - \frac{e^{i\phi}\gamma_c}{2\sqrt{2\pi}\sigma} \text{erfcx}\left(\frac{(\gamma/2) + i\Delta'}{2\sqrt{2\pi}\sigma}\right). \quad (1)$$

Here, $\Delta' = 2\pi(\mu - f_p)$ and $\text{erfcx}(\cdot)$ is the scaled complementary error function (see Supplementary Note 3). The magnitude of $S_{11}^{ave}$ is equivalent to the Voigt profile in laser spectroscopy[26]. Considering the contribution of the external circuitry, the real measured reflection coefficient is denoted as $S_{11}^{raw}$ (see Supplementary Note 5).

## Data availability

The data that support the findings of this study are provided in the paper. Source data are provided with this paper.

## Code availability

The codes for analyzing the data of this study are provided with this paper.

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

## Acknowledgements

We thank Matti Partanen and Kirill G. Fedorov for the preliminary characterization of the nanobolometer. This work is supported by the Academy of Finland Center of Excellence program (No. 336810), European Research Council under Advanced Grant ConceptQ (No. 101053801), Business Finland Foundation through Quantum Technologies Industrial (QuTI) project (No. 41419/31/2020), Technology Industries of Finland Centennial Foundation, Jane and Aatos Erkko Foundation through Future Makers program and SystemQ project, Finnish Foundation for Technology Promotion (No. 8640), and Horizon Europe program HORIZON-CL4-2022-QUANTUM-01-SGA via the project 101113946 OpenSuperQPlus100.

## Author contributions

A.K. and Q.C. performed the experiment and analyzed data. J.G. fabricated the nanobolometer. Q.C. developed the thermal radiation source. Q.C. and A.K. programmed FPGA and developed measurement software. A.G. contributed to measurement software. V.V. contribute to experimental setup. P.S., J.M., and Q.C. contributed to nanobolometer design and fabrication. Q.C., A.K., and M.M. wrote the manuscript with input from all authors. M.M. contributed to data analysis and supervised the project.

## Competing interests

M.M. is a co-founder and shareholder of IQM Finland Oy. M.M. is an inventor of patents FI122887B, US9255839B2, JP5973445B2, and EP2619813B1 titled "Detector of single microwave photons propagating in a guide". Other authors declare no competing interests.
