## [Transparent Peer Review file · Nature Communications]

Correlation measurement of propagating microwave photons at millikelvin

Corresponding Author: Dr Qi-Ming Chen

Version 0:

Reviewer comments:

Reviewer #1

(Remarks to the Author)

This is an interesting paper that should be published in its present form.

s_{11} appears to be listed only in arbitrary units, it would be useful to know what its typical values are.

What's really amazing is that such low power fields can be measured with such enormous drive fields, and that is why it would be interesting to know typical s_{11} values and these should be included.

100 attowatts is -150 dBm, so in some crude sense if s_{11} is -10 dB the thermometry represents an amplification of the noise field by 110 dB, when compared to the probe signal 30 dBm. This means that the noise level of the probe signal needs to be at this level, probably amplitude noise mainly for a broad signal of 113 MHz. This seems to be within the specs for a high quality signal generator for frequencies sufficiently separated from the carrier.

Of course, this is for a broad signal that fills the bandwidth. The narrow components of that signal can't be discriminated, of course, as this is a bolometric system.

I can see lots of uses for this system, so I am more interested in the technical details.

(Remarks on code availability)

even if there was code, I would not review it.

Reviewer #2

(Remarks to the Author)

The authors present a technique to measure the properties of microwave photons directly at cryogenic temperatures using a nanobolometer. This could avoid the need to amplify microwave photons to detectors at 300 K, a process which adds noise to the measurement. This technique could be useful as a tool for improving the readout of qubits, as well as any other studies using microwave photons - the authors suggest fundamental tests of quantum mechanics. In order to demonstrate the properties of this cryogenic detector the authors study the statistics of microwave photons generated by a blackbody and coherent source and compare the measurement to theoretical expectations. Because of the quality of this work and its application to both fundamental physics and quantum information, I recommend publication in Nature Communications following incorporation of a few comments and questions:

1. During the discussion of photon number variance, the authors mention that the fits for incoherent and coherent sources use almost identical fitting parameters. Referring to Extended Data Fig. 3d, the agreement between the different sources is excellent above approximately 510 MHz, however they deviate from each other below that point. If possible, it would be interesting for the authors to provide some possible explanations for this difference.

2. When comparing the frequency shift created by coherent and incoherent fields (Figure 3b), the authors apply a temperature-dependent power scaling as a correction to the incoherent data. Could the authors comment on the motivation for having the power scale linearly with the temperature, as opposed to some other scaling? For example, one might expect

$P \sim T^5$ due to electron-phonon coupling between the electron temperature in the thermometer and the phonon temperature in the substrate.

(Remarks on code availability)

Reviewer #3

(Remarks to the Author)

This manuscript presents a measurement technique based on a superconductor-normal metal-superconductor (SNS) junction nanobolometer. The authors present several interesting results based on reflection measurements of an SNS junction resonator at low temperatures (60-80mK). Key differences are shown between the photon statistics of thermal and coherent sources in microwave photons. However, there are several significant concerns.

1. The incident photon level is improperly normalized resulting in much larger incident photon levels than may be expected. Taking the input signal in Fig. 1d to be approximately -26dBm where a noticeable change in the center frequency occurs, we find the input signal is 2.51×10^{-14} W (assuming 80dB attenuation). Then the incident photon flux is $(2.51 \times 10^{-14} / 5.57 \times 10^{-24}) = 4.5 \times 10^9$ photons/s). It seems the authors are dividing this number by the center frequency, to get 0.5. The authors claim 0-24 photon/s/Hz input signal. This is correct, but confusing, because the number of photons/s/Hz is not relevant unless they can do single shot measurements with 0.5ns pulses. If we look at the photons/s, we see that there are significantly more photons present.

Next, note that the 500kHz low-pass filter averages over photon levels during a significantly longer integration time than a single shot at resonance of 8.4GHz. Later, the paper mentions that a 32us period is averaged over. If we take 32us to be the integration time, there are an average of 144,415 photons in a measurement. If we take 500kHz, we have 9026 photons in one measurement. Either way, this is hardly the shot noise limited regime.

2. Similar problems occur in Fig. 2. The actual values can be found by summing the Planck curve over the bandwidth of the resonator. Black body photons within the resonator bandwidth will also be on the scale of 1×10^9 photons/s at 1K. Therefore, it's not surprising that the frequency shift for a 1K black body is similar to the shift found for a -26dBm coherent source. This makes it difficult to say that the detector is photon number resolving as the average number of photons is significantly larger than the scale of quantized changes in photon number.

3. The extraction of average photon number $\langle n \rangle$ or photon variance $\langle \Delta n^2 \rangle$ is not clearly specified in the paper. Some detail is mentioned in the methods, and it seems that the authors want to relate the frequency shift to the average photon number, and the linewidth to the variance, but the authors do not specify how exactly photon number or photon variance are extracted from these measurements. Additionally, they mention that the extraction requires some fitting parameters. Four of these parameters differ between coherent and thermal sources, so it is difficult to tell how much of the changes are coming from differences in the reflection, and how much is coming from differences in the fitting parameters. Therefore, it is difficult to tell if the results for photon variance and $g^2(0)$ are theoretically rigorous.

4. The author mentions that the absorber increases the quasiparticle temperature of the nanowire, but the nanowire is a normal metal, and therefore quasiparticles should not exist in the nanowire, barring any interface effects.

There are also a few minor errors, such as missing words (millikelvin temperature in abstract), or typos (figure 1b caption, probe P_p and input P_h should be flipped). I would recommend significant revisions before publication.

(Remarks on code availability)

Reviewer #4

(Remarks to the Author)

(Remarks on code availability)

Version 1:

Reviewer comments:

Reviewer #1

(Remarks to the Author)

I've already reviewed this paper and said it should be published after the authors address my comments and those of the other referees. I'm not doing a second review, it is up to the paid editors to check that sort of thing.

(Remarks on code availability)

Even if I had received the code I would never trust anything written by anyone else. So this is something that is completely irrelevant to me. I also do not want to check the work at this level of detail.

Reviewer #2

(Remarks to the Author)

The authors have thoroughly answered all of my comments and questions, I now recommend the manuscript for publication. Congratulations on the exciting result!

(Remarks on code availability)

Reviewer #3

(Remarks to the Author)

Response to Reviewer

The authors have made significant additions to the supplementary which address my concerns on the fitting parameters. Furthermore, adding the source power to figures 2 and 3 and adding photon flux into the text helps to alleviate confusion on the scale of the input light. However, I still have concerns about the photon flux normalization and the single photon definition. The main concern is that the lower demonstrated sensitivity of the technique will limit potential applications. Due to the significantly reduced sensitivity of the presented technique, I cannot recommend this manuscript for publication in Nature Communications.

Below are my concerns on the flux normalization and single photon definition.

1. The normalization used in this manuscript for photon number is non-standard. In this manuscript the resonator bandwidth is used to normalize the input photon flux. In the rebuttal, reference 3 and 4 are cited for this normalization procedure. However, in reference 3 of the rebuttal a 400kHz "measurement bandwidth" which differs significantly from the resonator bandwidth (2-3MHz) is used as a normalization. In the same reference, the method to obtain the measurement bandwidth called Planck spectroscopy (i.e. an autocorrelation measurement) is cited [1]. In reference 4 of the rebuttal, the measurement bandwidth is given by Planck spectroscopy [1]. In both cases the bandwidth is limited by the measurement setup and not the resonance bandwidth. If we assume for this manuscript that the measurement bandwidth is limited by the integration time, we have at least 681 photons/s/Hz (for the reported 0.16 photons/s/Hz case). If we assume that it is limited by the 500kHz filter, then we have at least 43 photons/s/Hz. These values are significantly larger than the 0.1 photons/s/Hz found in reference 3 and 4 of the rebuttal.

2. Additionally, the argument for 'single photons' provided in the supplementary and rebuttal is not necessarily accurate even when accounting for the modes of the resonator. The authors are correct that resonator modes may offer single photon sensitivity, and that the sensitivity of these modes are the relevant metric in this experiment. However, in these measurements there are a significant number of photons per mode. Generally speaking, in photon correlations and photon statistics we can expect that the number of modes will be given by the integration time divided by the source coherence time (see Eq. 14.8-22 in Mandel and Wolf [2]). Therefore, there are 4256 modes in a 32 μ s measurement with 133MHz source bandwidth. Taking the photon flux over the number of modes and assuming equal distribution of photons per mode, we find that there are at least 5000 photons (for 0.16 photons/s/Hz case) in a single mode. Single photon sensitivity is possible if the authors choose a longer integration time, for example 6.25s. However, this would require selecting a filter of 0.16Hz, which may be difficult to realize in practice.

References:

[1] Mariantoni, M. et al. Planck Spectroscopy and Quantum Noise of Microwave Beam Splitters. Phys. Rev. Lett. 105, 133601 (2010).

[2] Mandel, L. & Wolf, E. Optical Coherence and Quantum Optics. (Cambridge University Press, Cambridge ; New York, 1995)

(Remarks on code availability)

I was able to install and run the code.

Version 2:

Reviewer comments:

Reviewer #3

(Remarks to the Author)

In the rebuttal, the authors compare their sensitivity to room temperature detectors which inherently have more dark noise. However, if we compare their device to other cryogenic detectors, we find several examples with better sensitivity [1-5]. Note that in these references, better sensitivity is demonstrated both for radiation coming from mK-scale black body sources, and from other coherent light sources properly normalized to photons/(s x Hz).

Additionally, the authors claim that the bandwidth is defined by the minimum bandwidth over which power is absorbed. It is true that the nanobolometer absorbs power over a larger bandwidth and could potentially have a larger measurement bandwidth. However, in the measurement presented by the authors, the nanobolometer response is sent through a series of

amplifiers and recorded by a room temperature detector where additional filtering is utilized. This additional filtering will limit the response of the detector to a new bandwidth dependent on the overall measurement setup. An analysis of the autocorrelations from a standard source could help the authors determine this bandwidth (see Planck spectroscopy [2]). The key point is that this bandwidth is dependent on the features of the setup which measures the nanobolometer response, not on the bandwidth over which power is absorbed.

The authors further argue that the aspects of their measurement setup are not relevant to photon correlation measurements. Specifically, they argue that the linewidth is not a time-dependent variable. However, their data in Fig. 2a directly contradicts this, as noise in the resonance frequency is clearly evident below the fitting. Additionally, correlation measurements are generally considered to be dependent on the measurement bandwidth (see [6] eq. 14.8-22). This is why other techniques [3,4] have characterized their measurement bandwidth using autocorrelation techniques such as Planck spectroscopy, and why the initial paper on microwave Planck spectroscopy cites [6].

Lastly, the authors mention the units of photons/(s x Hz). I do not argue with the usage of this unit of photons/(s x Hz) in general. However, calling these quantities 'single photons' is misleading. Photon correlations can expand beyond a single Hz depending on the measurement setup and integration time. Since the measurement modes may be larger than a single Hz, the modes may contain many photons while still having normalized photon levels on the order of 1 photons/(s x Hz). For example, there may be 31250 photons/s in a 31.25kHz measurement bandwidth with 4256 modes (i.e. 32us integration time with 133MHz source bandwidth), this would lead to 1 photons/(s x Hz), however there would be 7 photons in each mode.

Due to the uncorrected normalization of input light and the lack of demonstrated sensitivity I do not recommend this manuscript for publication in Nature Communications.

[1] Renger, M. et al. Beyond the standard quantum limit for parametric amplification of broadband signals. npj Quantum Inf 7, 1–7 (2021).

[2] Mariani, M. et al. Planck Spectroscopy and Quantum Noise of Microwave Beam Splitters. Phys. Rev. Lett. 105, 133601 (2010).

[3] Goetz, J. et al. Photon Statistics of Propagating Thermal Microwaves. Phys. Rev. Lett. 118, 103602 (2017).

[4] Fedorov, K. G. et al. Experimental quantum teleportation of propagating microwaves. Science Advances 7, eabk0891 (2021).

[5] Gandorfer, S. et al. Two-dimensional Planck spectroscopy for microwave photon calibration. arXiv:2308.02389 [quant-ph].

[6] Mandel, L. & Wolf, E. Optical Coherence and Quantum Optics. (Cambridge University Press, Cambridge ; New York, 1995).

(Remarks on code availability)

The code seems to run.

Version 3:

Reviewer comments:

Reviewer #4

(Remarks to the Author)

The authors have responded to many of the questions I have had about the work and about its importance. After considering the authors' response and modifications, I recommend this work for publication after some minor revisions.

After further consideration, I now agree with the authors that the presented technique is fundamentally different from low temperature parametric amplification techniques. As the authors mentioned, the frequency of detection and measurement are completely different. To help readers understand the advancement, the authors should better explain in the manuscript how the work differs from these prior techniques. For example, they could highlight the difference in probe and detection frequencies.

As the authors mention in their reply, the photon correlation is a key point of novelty in this work. I still believe some additional analysis needs to be done to demonstrate that the probe and detection processes are actually independent. From the presented experiments I find it difficult to tell if this is actually the case. I understand that there is no electrical coupling between the probe field and the input field, however in single photon detection experiments with 1550nm light, the single photon detectors can be described in a similar manner. In that case, the detector such as a superconducting nanowire single photon detector acts as a bolometer, and produces a GHz signal from incident 1550nm light.

Additionally, in many photon correlation experiments the readout and detection processes are not independent, and this reflects in the g_2 correlations. In experiments with 1550nm photons it has been demonstrated that by increasing the measurement time the measured distribution from a thermal source becomes more Poissonian (see [1] Fig. 3f) and therefore g_2 approaches 1. In the manuscript, deviations from theoretically predicted g_2 is observed in Fig. 4, especially in cases with high levels of thermal photons. These deviations do not necessarily demonstrate the probe and detection processes are dependent, but I believe further investigation is necessary. To test this, the authors could verify that when decreasing the FIR filter bandwidth, and when increasing the integration time, the g_2 correlations from a thermal source do not significantly change.

If these g_2 correlations do indeed depend on the FIR filter bandwidth or integration time, then the deviation from theory can be accounted for by treating the photon number as a negative binomial distribution [1-2]. For a negative binomial distribution, the photon number variance will go by $\langle n \rangle + \langle n \rangle^2/M$ where M is the number of modes with $M = \tau \cdot BW$ [3], where τ is the measurement time and BW is the absorbed BW of thermal light. By fitting the measured g_2 values to this distribution, they can find M and therefore τ . Then, $1/\tau$ can replace the FWHM mentioned in Supplementary section 1, and the input light can be properly normalized with this updated value.

[1] Cheng, R. et al. A 100-pixel photon-number-resolving detector unveiling photon statistics. Nat. Photon. 17, 112–119 (2023).

[2] Katamadze, K. G., Avosopiants, G. V., Bogdanova, N. A., Bogdanov, Yu. I. & Kulik, S. P. Multimode thermal states with multiphoton subtraction: Study of the photon-number distribution in the selected subsystem. Phys. Rev. A 101, 013811 (2020).

[3] Mandel, L. & Wolf, E. Optical Coherence and Quantum Optics. (Cambridge University Press, Cambridge ; New York, 1995).

(Remarks on code availability)

The code seems to run

Version 4:

Reviewer comments:

Reviewer #3

(Remarks to the Author)

The authors have addressed all of my comments. I recommend the manuscript for publication.

(Remarks on code availability)

The code seems to run.

NCOMMS-24-47168-T – Reply to Reviewer #1

This is an interesting paper that should be published in its present form.

Reply: We are grateful for the appreciation of our work and the recommendation for publication.

S_{11} appears to be listed only in arbitrary units, it would be useful to know what its typical values are.

What's really amazing is that such low power fields can be measured with such enormous drive fields, and that is why it would be interesting to know typical S_{11} values and these should be included.

Reply: In the revised manuscript, we relabeled the heat and probe powers, P_h and P_p , to the sample input that provide more direct information of the field intensity at the sample stage. Since the actual attenuation of the probe line is not characterized in the experiment, we sum over the attenuation of the attenuators and the directional coupler, 80 dB, but neglect any cable loss from the source to the sample.

In the revised manuscript, we re-express $|S_{11}^{\text{raw}}|$ of Fig. 1 in dB scale. Here, we define $|S_{11}^{\text{raw}}|$ at the highest measurement frequency, i.e., 530 MHz, as 0 dB. Because of the asymmetry lineshape at high probe power and the unfavorable Lorentzian background as shown in Supplement Fig. 1 in the revised manuscript (it is Extended Data Fig. 3 in the initial submission), the maximum value of $|S_{11}^{\text{raw}}|$ can slightly overshoot 0 dB (maximum 1.6 dB in Fig. 1a, nominally 0.15 dB in Figs. 1b and 1c). The minimum value of $|S_{11}^{\text{raw}}|$ is nominally -3.8 dB in Figs. 1b and 1c, while it is -5.3 dB in Fig. 1a.

In the revised manuscript, we did not re-express $|S_{11}^{\text{raw}}|$ of the Supplement Fig. 1 in dB scale for two reasons: (i) Here, we measure $|S_{11}^{\text{raw}}|$ in a large frequency range that contains a background peak. Normalizing $|S_{11}^{\text{raw}}|$ at 540 MHz may cause confusions of the large overshoot around this background peak (500 MHz). (ii) This measurement is performed on our FPGA setup which is different from the vector network analyzer (VNA) measurement in Fig. 1. Here, the y-axis value is in fact the voltage we measured in the unit of mV. However, we could also change the y-axis in dB scale if the referee thinks that it is a better way of expressing the data.

100 attowatts is -150 dBm, so in some crude sense if S_{11} is -10 dB the thermometry represents an amplification of the noise field by 110 dB, when compared to the probe signal 30 dBm. This means that the noise level of the probe signal needs to be at this level, probably amplitude noise mainly for a broad signal of 113 MHz. This seems to be within

the specs for a high quality signal generator for frequencies sufficiently separated from the carrier.

Of course, this is for a broad signal that fills the bandwidth. The narrow components of that signal can't be discriminated, of course, as this is a bolometric system.

I can see lots of uses for this system, so I am more interested in the technical details.

Reply: This is an insightful comment and let us provide the reply by starting with why we design the nanobolometer as it is.

Generally speaking, we minimize the thermal conductance and heat capacity of the nanowire such that a tiny radiation can cause a significant electron temperature change for thermometry. This is so far the normal routine of designing a hot-electron bolometer. The key difference beyond the conventional design is the thermometer part: We use an array of short SNS junctions to sense this temperature change, and embedded them into a sub-gigahertz resonator with low impedance. Here, the fragile process of Andreev reflection could be very sensitive to a tiny temperature change, and lead to an even more noticeable change in the inductance, $N\delta L$, where N is the number of junctions. The low impedance resonator may further amplify this change as its resonance shifts as $1/[C(L + N\delta L)]$, where C and L are the capacitance and inductance of the thermometer before perturbation. Moreover, the final signal is read out from the resonance shift of the thermometer. Thanks to the easily accessible atomic frequency standard, frequency shift may be the most precise measurement one could perform in a normal laboratory.

In specific to Figs. 1 and 2, we use the vector network analyzer (VNA, Rohde & Schwarz ZNB 40) to get the data. At each probe frequency, we set the intermediate-frequency (IF) bandwidth to 200 Hz with 10 averages for one data point. The frequency is stabilized to an Rb frequency standard (SRS FS725), which has less than -130 dBc/Hz phase noise from the data sheet. In our experiment, an 100 aW signal would already cause a frequency shift at the level of 100 kHz, which is orders of magnitude larger than the frequency resolution we could have. Indeed, there is still a plenty of space to improve the power resolution, by averaging out the power uncertainty of our source or choosing a smaller IF bandwidth, before the devices may limit us. The limit is possibly at the 1 Hz level, which is limited earlier by the so far best achievable Q factor of a sub-gigahertz resonator before reaching the limit of the measurement instruments we use.

(Remarks on code availability):

even if there was code, I would not review it.

NCOMMS-24-47168-T – Reply to Reviewer #2

The authors present a technique to measure the properties of microwave photons directly at cryogenic temperatures using a nanobolometer. This could avoid the need to amplify microwave photons to detectors at 300 K, a process which adds noise to the measurement. This technique could be useful as a tool for improving the readout of qubits, as well as any other studies using microwave photons - the authors suggest fundamental tests of quantum mechanics. In order to demonstrate the properties of this cryogenic detector the authors study the statistics of microwave photons generated by a blackbody and coherent source and compare the measurement to theoretical expectations. Because of the quality of this work and its application to both fundamental physics and quantum information, I recommend publication in Nature Communications following incorporation of a few comments and questions:

Reply: We appreciate that our work is perceived this way and the recommendation for publication after properly replying to the following questions.

1. During the discussion of photon number variance, the authors mention that the fits for incoherent and coherent sources use almost identical fitting parameters. Referring to Extended Data Fig. 3d, the agreement between the different sources is excellent above approximately 510 MHz, however they deviate from each other below that point. If possible, it would be interesting for the authors to provide some possible explanations for this difference.

Reply: We first explain how we end up with the fitting parameters as shown in the Supplement Fig.1 in the revised manuscript (it is Extended Data Fig.3 in the initial submission). In the experiment, we set the same initial guess for all the measurements of this study, which are fitting results at the lowest power and lowest temperature. We also use the same optimizer, i.e., the “least_squares” function in Scipy package, for all the fitting processes. The goal is to fit the measured spectrum in the complex domain with no any additional information that can bias the optimizer. With that being said, the clear distinction of $g^{(2)}(0)$ between coherent and incoherent inputs is an experimental observation instead of our choice. In the end, we plot the fitting results of 14 independent coherent-input measurements and 20 independent incoherent-input measurements together in the Supplement Fig.1. We see that the coherent and incoherent data show good self consistency within each data set, and the two data sets differ by only a few percentage at the maximum. In specific, relative deviations of γ_c , ϕ , f_b , and ϕ_b are 6.03 %, 11.87 %, 0.08 %, and 2.92 %, respectively.

We now try to answer the question that why the fitting parameters start deviating from

each other below 510 MHz. One indication from the wide-frequency sweep (Supplement Fig. 1) is that the deviation emerges when resonance frequency gets closer to the spurious background peak. In our measurement, we observe a shift of this background signal when changing the mixing chamber (MXC) temperature, T_b . This is indeed the reason we set the background peak frequency, f_b , as a fitting parameter. A large mismatch between the sample output and the amplifier is suspected from these observations. Nevertheless, we may argue that the difference of 2/4 of the different fitting parameters are the background-signal properties, i.e., the frequency and phase offset of the background peak, f_b and ϕ_b . The 3rd one is the phase offset of the bolometer response, ϕ , which is related to the suspected impedance mismatch. The last parameter, γ_c , that characterizes the nanobolometer itself shows merely $0.77 \mu\text{s}^{-1}$ difference at the averaged base level of $12.37 \mu\text{s}^{-1}$.

In these regards, we add the following explanation of the small difference in Supplementary Note 4 (paragraph 3):

“The fitted results are almost identical for thermal and coherent inputs with maximally few percentage of difference at the highest temperatures ($T > 1.7 \text{ K}$), which we attribute to the changing of the bath temperature T_b as measured in Fig. 3b of the main text.”

Finally, as the reviewer #3 raised a concern on the sensitivity of the correlation result, $g^{(2)}(0)$, with regard to the fitting parameters, we asked ourselves what if we force the fitting parameters to be the same. Here, we exchange the fitting parameters of the coherent-input (thermal-input) data with the thermal-input (coherent-input) fitting parameters and extract $g^{(2)}(0)$. We take the data at $f_r \simeq 507 \text{ MHz}$ for example, as it is the largest deviation point. We use the least square method to extract the resonance frequency, μ , and the broadening, σ^2 . We observe that the extracted value of $g^{(2)}(0)$ is robust to the specific choices of the parameter sets, as shown in Fig. 1 of this rebuttal letter. In the revised manuscript, we keep the parameter deviation as it is in order not to intervene the optimization process described above. This is in the end a matter of choice. We could also manually force the parameters to be the same and update the results in Figs. 3 and 4.

2. When comparing the frequency shift created by coherent and incoherent fields (Figure 3b), the authors apply a temperature-dependent power scaling as a correction to the incoherent data. Could the authors comment on the motivation for having the power scale linearly with the temperature, as opposed to some other scaling? For example, one might expect $P \sim T^5$ due to electron-phonon coupling between the electron temperature in the thermometer and the phonon temperature in the substrate.

Reply: The referee is correct that the cooling power is a n th-order polynomial of T_b , where $n = 5$ for normal metal in the clean limit. In our device, we observe $n = 6$ as it is also

Figure 1: **a-b** Stability test of the fitting parameters for coherent and thermal inputs at $f_r \simeq 507$ MHz. In all the panels, the dots are the raw data, and the red curves are originally fitted data as shown in Supplement Fig. 1. The blue curves represent the coherent (thermal) data but with thermal (coherent) fitting parameters, where we use the least square method to extract the resonance frequency, μ , and the broadening, σ^2 , that lead to the best fit of the raw data.

observed for disordered metal in the dirty limit [1]. The detailed characterization of the thermal properties is performed in a separate work in preparation.

In the initial submission, we use a linear relation to avoid detailed discussions of the thermodynamics, especially the value of the order parameter n . Considering a small perturbation of the phonon temperature, $T_b \rightarrow T_b + \delta T_b$, the additional cooling power is $-G_{\text{eb}}\delta T_b$, where $G_{\text{eb}} = \Sigma V T_b^{n-1}$ is the thermal conductance at temperature T_b with ΣV being a constant. The previously defined scaling factor, β , is basically the averaged value of G_{eb} over the relevant range of T_b .

Since the linear approximation seems to confuse the real thermodynamics, we replaced Fig. 3b with n th-order polynomial contribution of the phonons in the revised manuscript. The additional power due to phonon temperature change is $\beta [(T_b + \delta T_b)^n - T_b^n]$ with $n = 6$, while we neglect the latter item for simplicity. Here, the physical meaning of β is ΣV , as it is different from the initial submission. We add the plot here for self consistency (Fig. 2a of this rebuttal letter).

Figure 2: **a** Replacement of Fig. 3 in the revised manuscript, where we have assumed an 6th-order correction of the phonon temperature. **b** The same result but assuming a non-unitary absorption rate $\kappa = 0.5$, where the photon number is the value seen by the absorber. It is worth noting that the scaling factors for phonon-temperature corrections are different because of the different choices of κ . To get (b), we need to resale every parameter defined in the “scaling_factors.nc” file by κ .

On additional comment on the new plot: The result with n th-order correction still does not reproduce the same curve as with a coherent input. This motivates us to investigate on the absorption rate, κ , i.e., impedance mismatch at the absorber side, which was neglected throughout this study. We found a good overlap for the two curves at $\kappa \simeq 0.5$, as shown in Fig. 2b of this rebuttal letter. It indicates that the absorber is not perfectly matched to the input line, which may not be surprising. We could also relabel the photon number and variance in Figs. 3 and 4 by rescaling all the parameters stored in the “scaling_factors.nc” file by κ . Here, photon number will denote the real absorbed value instead of the input value. However, we did not do this rescaling because we think the non-unitary κ is the property of the nanobolometer itself. The rescaling may lead to an exaggeration of its real performance. Instead, we add the following sentence in line 181:

“A quantitative agreement can be achieved by assuming an approximately 50% absorption rate of the nanobolometer, which is not taken into account throughout this study.”

NCOMMS-24-47168-T – Reply to Reviewer #3

This manuscript presents a measurement technique based on a superconductor-normal metal-superconductor (SNS) junction nanobolometer. The authors present several interesting results based on reflection measurements of an SNS junction resonator at low temperatures (60-80 mK). Key differences are shown between the photon statistics of thermal and coherent sources in microwave photons. However, there are several significant concerns.

Reply: We of course appreciate that our work is perceived as "interesting" and the careful proofreading by the referee.

Indeed, we agree with the "significant concerns" of the referee – the photon flux is significantly larger than 1 photon/s while the chosen unit of photons/(s·Hz) makes it sound small. However, we argue that the choice of unit depends on the purpose of this study. Here, we are interested in the correlation of photons in a continuous beam of radiation instead of correlations of individual photon emission events. The former is much easier and can date back to the celebrated Hanbury Brown & Twiss experiment, while the latter is achieved only recently in the optical regime [2].

In the following reply, we will explain why the unit of single photons has to be photon/(s·Hz) for our interest. This is also the conventional choice of unit for similar studies in the literature. For example, measurement of propagating photon correlations with linear detectors [3] and quantum teleportation with propagating microwaves [4]. More detailed derivations and the corresponding references can be found in the Supplementary Note 1–4.

1. The incident photon level is improperly normalized resulting in much larger incident photon levels than may be expected. Taking the input signal in Fig. 1d to be approximately –26 dBm where a noticeable change in the center frequency occurs, we find the input signal is 2.51×10^{-14} W (assuming 80 dB attenuation). Then the incident photon flux is $(2.51 \times 10^{-14} / 5.57 \times 10^{-24}) = 4.5 \times 10^9$ photons/s). It seems the authors are dividing this number by the center frequency, to get 0.5. The authors claim 0-24 photon/s/Hz input signal. This is correct, but confusing, because the number of photons/s/Hz is not relevant unless they can do single shot measurements with 0.5 ns pulses. If we look at the photons/s, we see that there are significantly more photons present.

Reply: We now explain why the unit of single photons has to be photons/(s·Hz), in order to measure the correlations of propagating photons in a stationary beam of radiation.

Photons are excitations of electromagnetic waves that has multitude of degrees of freedom. The concept of single photons relies highly on the specific definition of modes. In cases

of cavity quantum electrodynamics, it is convenient to define modes as those standing waves confined in a finite volume. The corresponding quantity, $\hbar\omega_k a_k^\dagger a_k$, is the energy of excitation of the k th mode. However, this definition will soon encounter problems when discussing a propagating field. For example, the 2nd-order correlation function, $g^{(2)}(\tau)$, is ill defined if we stick to this unit, as it is already noticed when explaining the Hanbury Brown & Twiss experiment [5]. A more suitable choice is $a_f(t) = \int_{-\infty}^{+\infty} f^*(\omega) a(\omega) e^{-i\omega t} dt$, where $[a(\omega), a^\dagger(\omega')] = \delta(\omega - \omega')$ and $\int_{-\infty}^{+\infty} |f(\omega)|^2 = 2\pi$. We have $[a_f(t), a_f^\dagger(t)] = 1$. This is the basis on which we define “single photons” in a propagating field. One can verify that the unit of $a_f^\dagger(t) a_f(t)$ is photons/(s · Hz). The detailed calculations are shown in the Supplementary Note 1.

The above derivation explains how we convert the power to the photon number. For thermal fields, we get the power per frequency from the Planck’s law, and multiply it by $1/(\hbar\omega)$ to get the photon number in the unit of photons/(s · Hz). For coherent fields, we multiply the input power by $1/(\text{FWHM}\hbar\omega)$, assuming that the noise level of the source is significantly smaller than the power level.

To avoid ambiguity that the readers may have, we add the following sentences in line 137:

“, corresponding to a considerable amount of individual photon wavepackets per second, 2.1×10^7 photons/s. In this regard, the nanobolometer reported here is still far from resolving each individual photon mission events in a microwave field, which has been demonstrated in the optical regime [31]. Fortunately, the observed power resolution is adequate for resolving the quantum statistical properties of propagating microwave photons that are conventionally defined by the arriving photons per second per hertz [27] (see Supplementary Note 1).”.

Next, note that the 500 kHz low-pass filter averages over photon levels during a significantly longer integration time than a single shot at resonance of 8.4 GHz. Later, the paper mentions that a 32 μs period is averaged over. If we take 32 μs to be the integration time, there are an average of 144,415 photons in a measurement. If we take 500 kHz, we have 9026 photons in one measurement. Either way, this is hardly the shot noise limited regime.

Reply: We believe that the concern on the unit of single photons is answered by reply #1. Here, we would like to make a further clarification on the integration time. More detailed discussions can be found in the Supplementary Note 2 and 3.

Basically, the correlation measurement is performed by the bolometer itself and we apply a probe field only for reading out the averaged measurement result, i.e., the spectrum broadening. The argument is the central limit theorem, where the variance of the electron temperature is proportional to that of the input power (Supplementary Note 2). The

former is readout from the averaged spectrum through the averaged line-width broadening (Supplementary Note 3). The 500 kHz filter and 32 us average is to improve the signal to ratio, which does not influence the behaviour of the electron temperature.

2. Similar problems occur in Fig. 2. The actual values can be found by summing the Planck curve over the bandwidth of the resonator. Black body photons within the resonator bandwidth will also be on the scale of 1×10^9 photons/s at 1 K. Therefore, it's not surprising that the frequency shift for a 1 K black body is similar to the shift found for a -26 dBm coherent source. This makes it difficult to say that the detector is photon number resolving as the average number of photons is significantly larger than the scale of quantized changes in photon number.

Reply: We believe that the concern on the unit of single photons is answered by reply #1. For a stationary beam of radiation, the proper unit of single photons is photons/(s · Hz) that is different from the unit for a standing wave. This is the unit where we can expect quantum statistical properties of radiation.

In the revised manuscript, we label the power in line with the photon number in Figs. 2-4 to avoid the confusion on the power resolution of our nanobolometer.

3. The extraction of average photon number $\langle n \rangle$ or photon variance is not clearly specified in the paper. Some detail is mentioned in the methods, and it seems that the authors want to relate the frequency shift to the average photon number, and the linewidth to the variance, but the authors do not specify how exactly photon number or photon variance are extracted from these measurements.

Reply: We made the following changes in the revised manuscript to explain our signal processing procedure in more detail:

We add the definition of photon number in line 114: “ $\langle \hat{n} \rangle = P_h / (\text{FWHM} h f_0)$ ”.

We add the equivalent definition of photon number specific to thermal input in line 154: “ $\langle \hat{n} \rangle$ is precisely defined by the Planck's law, ...”

We add Supplementary Note 4 to provide step-by-step explanation of our data processing procedure.

Additionally, they mention that the extraction requires some fitting parameters. Four of these parameters differ between coherent and thermal sources, so it is difficult to tell how much of the changes are coming from differences in the reflection, and how much is coming from differences in the fitting parameters. Therefore, it is difficult to tell if the results for

photon variance and $g^{(2)}(0)$ are theoretically rigorous.

Reply: A detailed reply regarding the fitting parameters is provided below comment 1 of reviewer #2. We duplicate it here for self consistency:

We first explain how we end up with the fitting parameters as shown in the Supplement Fig. 1 in the revised manuscript (it is Extended Data Fig. 3 in the initial submission). In the experiment, we set the same initial guess for all the measurements of this study, which are fitting results at the lowest power and lowest temperature. We also use the same optimizer, i.e., the “least_squares” function in Scipy package, for all the fitting processes. The goal is to fit the measured spectrum in the complex domain with no any additional information that can bias the optimizer. With that being said, the clear distinction of $g^{(2)}(0)$ between coherent and incoherent inputs is an experimental observation instead of our choice. In the end, we plot the fitting results of 14 independent coherent-input measurements and 20 independent incoherent-input measurements together in the Supplement Fig. 1. We see that the coherent and incoherent data show good self consistency within each data set, and the two data sets differ by only a few percentage at the maximum. In specific, relative deviations of γ_c , ϕ , f_b , and ϕ_b are 6.03 %, 11.87 %, 0.08 %, and 2.92 %, respectively.

We now try to answer the question that why the fitting parameters start deviating from each other below 510 MHz. One indication from the wide-frequency sweep (Supplement Fig. 1) is that the deviation emerges when resonance frequency gets closer to the spurious background peak. In our measurement, we observe a shift of this background signal when changing the mixing chamber (MXC) temperature, T_b . This is indeed the reason we set the background peak frequency, f_b , as a fitting parameter. A large mismatch between the sample output and the amplifier is suspected from these observations. Nevertheless, we may argue that the difference of 2/4 of the different fitting parameters are the background-signal properties, i.e., the frequency and phase offset of the background peak, f_b and ϕ_b . The 3rd one is the phase offset of the bolometer response, ϕ , which is related to the suspected impedance mismatch. The last parameter, γ_c , that characterizes the nanobolometer itself shows merely $0.77 \mu\text{s}^{-1}$ difference at the averaged base level of $12.37 \mu\text{s}^{-1}$.

In these regards, we add the following explanation of the small difference in Supplementary Note 4 (paragraph 3):

“The fitted results are almost identical for thermal and coherent inputs with maximally few percentage of difference at the highest temperatures ($T > 1.7$ K), which we attribute to the changing of the bath temperature T_b as measured in Fig. 3b of the main text.”

Finally, as the reviewer #3 raised a concern on the sensitivity of the correlation result,

Figure 3: **a-b** Stability test of the fitting parameters for coherent and thermal inputs at $f_r \simeq 507$ MHz. In all the panels, the dots are the raw data, and the red curves are originally fitted data as shown in Supplement Fig. 1. The blue curves represent the coherent (thermal) data but with thermal (coherent) fitting parameters, where we use the least square method to extract the resonance frequency, μ , and the broadening, σ^2 , that lead to the best fit of the raw data.

$g^{(2)}(0)$, with regard to the fitting parameters, we asked ourselves what if we force the fitting parameters to be the same. Here, we exchange the fitting parameters of the coherent-input (thermal-input) data with the thermal-input (coherent-input) fitting parameters and extract $g^{(2)}(0)$. We take the data at $f_r \simeq 507$ MHz for example, as it is the largest deviation point. We use the least square method to extract the resonance frequency, μ , and the broadening, σ^2 . We observe that the extracted value of $g^{(2)}(0)$ is robust to the specific choices of the parameter sets, as shown in Fig. 3 of this rebuttal letter. In the revised manuscript, we keep the parameter deviation as it is in order not to intervene the optimization process described above. This is in the end a matter of choice. We could also manually force the parameters to be the same and update the results in Figs. 3 and 4.

4. The author mentions that the absorber increases the quasiparticle temperature of the nanowire, but the nanowire is a normal metal, and therefore quasiparticles should not exist in the nanowire, barring any interface effects.

Reply: The referee is correct that we normally do not call electrons and holes as

quasiparticles in a normal metal. However, the array of short SNS junctions in thermometer part makes the nanowire more than a pure piece of normal-metal. We probe the hot electrons by diffusing them into an junction array. Here, the “interface effects” happen and give rise to the concept of quasiparticles.

With that being said, we also agree with the referee that it is not necessary to introduce the concept of quasiparticles. In the revised manuscript, we removed the sentence “, or more precisely quasiparticle”.

There are also a few minor errors, such as missing words (millikelvin temperature in abstract), or typos (figure 1b caption, probe P_p and input P_h should be flipped). I would recommend significant revisions before publication.

Reply: We are grateful for the careful proofreading by the referee. We made the corrections and carefully proofread the manuscripts before resubmission.

NCOMMS-24-47168-T – Reply to Reviewer #4

References

- [1] Jian Wei, David Olaya, Boris S. Karasik, Sergey V. Pereverzev, Andrei V. Sergeev, and Michael E. Gershenson. Ultrasensitive hot-electron nanobolometers for terahertz astrophysics. *Nat. Nanotechnol.*, 3(8):496–500, July 2008.
- [2] J. Wiersig, C. Gies, F. Jahnke, M. Aßmann, T. Berstermann, M. Bayer, C. Kistner, S. Reitzenstein, C. Schneider, S. Höfling, A. Forchel, C. Kruse, J. Kalden, and D. Hommel. Direct observation of correlations between individual photon emission events of a microcavity laser. *Nature*, 460(7252):245–249, jul 2009.
- [3] J. Goetz, S. Pogorzalek, F. Deppe, K. G. Fedorov, P. Eder, M. Fischer, F. Wulschner, E. Xie, A. Marx, and R. Gross. Photon statistics of propagating thermal microwaves. *Phys. Rev. Lett.*, 118:103602, Mar 2017.
- [4] Kirill G. Fedorov, Michael Renger, Stefan Pogorzalek, Roberto Di Candia, Qiming Chen, Yuki Nojiri, Kunihiro Inomata, Yasunobu Nakamura, Matti Partanen, Achim Marx, Rudolf Gross, and Frank Deppe. Experimental quantum teleportation of propagating microwaves. *Sci. Adv.*, 7(52), December 2021.
- [5] K. J. Blow, Rodney Loudon, Simon J. D. Phoenix, and T. J. Shepherd. Continuum fields in quantum optics. *Phys. Rev. A*, 42:4102–4114, Oct 1990.

NCOMMS-24-47168-A – Reply to Reviewer #3

The authors have made significant additions to the supplementary which address my concerns on the fitting parameters. Furthermore, adding the source power to figures 2 and 3 and adding photon flux into the text helps to alleviate confusion on the scale of the input light. However, I still have concerns about the photon flux normalization and the single photon definition. The main concern is that the lower demonstrated sensitivity of the technique will limit potential applications. Due to the significantly reduced sensitivity of the presented technique, I cannot recommend this manuscript for publication in Nature Communications.

Reply: We are happy to see that the revised manuscript has removed the concerns on the soundness of our result. We now reply to the expressed main concern of the potential significance. That is, whether our cryogenic detector is less sensitive than a room-temperature detector.

For a quantitative comparison, we take the National Instrument 5782 as an example of the room-temperature detectors. Its voltage resolution is 0.125 mV according to the data sheet, corresponding to a power resolution of 0.3 nW for an 50Ω load. In comparison, our nanobolometer reports a power resolution of 0.12 fW, which is 6 orders of magnitude more sensitive than a nominal room-temperature detector. In this regard, we cannot agree that our nanobolometer has a “significantly reduced sensitivity” compared with the state of the art.

We also note that the here reported 0.1 fW resolution is a nominal figure of merit of our nanobolometers fabricated so far. We have achieved 0.01 fW and 0.001 fW power resolution in samples with superconductor-normal metal-superconductor (SNS) [Phys. Rev. Lett. 117, 030802 (2016)] and superconductor-graphene-superconductor (SGS) junctions [Nature 586, 47-51 (2020)], which have even higher improvement of sensitivity than the best room-temperature detectors.

Below are my concerns on the flux normalization and single photon definition.

1. The normalization used in this manuscript for photon number is non-standard. In this manuscript the resonator bandwidth is used to normalize the input photon flux. In the rebuttal, reference 3 and 4 are cited for this normalization procedure. However, in reference 3 of the rebuttal a 400kHz ‘measurement bandwidth’ which differs significantly from the resonator bandwidth (2-3MHz) is used as a normalization. In the same reference, the method to obtain the measurement bandwidth called Planck spectroscopy (i.e. an autocorrelation measurement) is cited [1]. In reference 4 of the rebuttal, the measurement bandwidth is given by Planck spectroscopy [1]. In both cases the bandwidth is limited by the measurement setup and not the resonance bandwidth. If we assume for this manuscript that the measurement bandwidth is limited by the integration time, we have at least 681 photons/s/Hz (for the reported 0.16 photons/s/Hz case). If we assume that it is limited by the 500kHz filter, then we have at least 43 photons/s/Hz. These values are significantly larger than the 0.1 photons/s/Hz found in reference 3 and 4 of the rebuttal.

Reply: To the best of our understanding, here the word “non-standard” indicates the different values of bandwidth compared with references 3 and 4. By definition, the bandwidth is the minimum bandwidth between the source and the detector, which determines the power absorbed by the detector.

In references 3 and 4, a chain of amplifiers are used to meet the threshold of sensitivity of room-temperature detectors. Because amplification necessarily introduces noise, the measurement bandwidth in such conventional setup should be chosen as small as possible to improve the signal to noise ratio. We therefore used a ~ 100 kHz-bandwidth digital filter before IQ sampling to compromise the latency and the FPGA resources. The Planck spectroscopy experiment is not used to determine this bandwidth.

The here reported cryogenic nanobolometer is free of amplification process for correlation measurement and is broadband in nature. In this study, we insert an 133 MHz-bandwidth coplanar waveguide filter just before the detector. This is the minimum bandwidth between the source and the cryogenic detector that defines the power to be absorbed.

We now explain further why the $32 \mu\text{s}$ integration time and the 500 kHz digital filter is not relevant to the correlation measurement. The photon detection process is implemented by the cryogenic detector itself, where the information of photon correlation is reflected by the linewidth of its readout resonator. The $32 \mu\text{s}$ integration time and the 500 kHz digital filter are parameters we use to readout the linewidth. The linewidth itself is not a time-dependent variable and does not depend on how fast we measure it. Thus, the input photon correlation should not depend on these parameters either. We hope that already this clarification answers most of the questions of the reviewer.

Finally, we emphasize the difference between the photon number as referred to the source and that absorbed by the detector. In a conventional setup with amplifiers and room-temperature detectors, there is nominally more than 80 dB amplification between the source and the detector. Thus, the real photon flux absorbed by the detector is approximately 10^{12} photons/s (assuming 0.1 photons/(s \times Hz) resolution, and 100 kHz bandwidth). In comparison, the here reported cryogenic detector is free of amplification and the real photon number absorbed by the detector is 10^7 photons/s (assuming 0.1 photons/(s \times Hz) resolution, and 100 MHz bandwidth). There are orders of magnitude improvement in the sensitivity when using the cryogenic detector. If we consider the amplification noise of the amplifier, the improvement will be larger even the amplifier is perfect.

2. Additionally, the argument for ‘single photons’ provided in the supplementary and rebuttal is not necessarily accurate even when accounting for the modes of the resonator. The authors are correct that resonator modes may offer single photon sensitivity, and that the sensitivity of these modes are the relevant metric in this experiment. However, in these measurements there are a significant number of photons per mode. Generally speaking, in photon correlations and photon statistics we can expect that the number of modes will be given by the integration time divided by the source coherence time (see Eq. 14.8-22 in Mandel and Wolf [2]). Therefore, there are 4256 modes in a 32us measurement with 133MHz source bandwidth. Taking the photon flux over the number of modes and assuming equal

distribution of photons per mode, we find that there are at least 5000 photons (for 0.16 photons/s/Hz case) in a single mode. Single photon sensitivity is possible if the authors choose a longer integration time, for example 6.25s. However, this would require selecting a filter of 0.16Hz, which may be difficult to realize in practice.

References:

- [1] Mariani, M. et al. Planck Spectroscopy and Quantum Noise of Microwave Beam Splitters. Phys. Rev. Lett. 105, 133601 (2010).
- [2] Mandel, L. & Wolf, E. Optical Coherence and Quantum Optics. (Cambridge University Press, Cambridge; New York, 1995)

Reply: We have thoroughly explained in our previous reply and in the Supplementary Notes why photons should be defined in the photon/(s × Hz) unit for a continuous beam of radiation. In addition, we have provided several references to support this point, not only by ourselves and collaborators [PRL 118, 103602 (2017); Sci. Adv. 7, 52 (2021)] but also by researchers in the optical regime [PRL 98, 153603 (2007)].

Here, we provide a popular textbook reference to strengthen this point. We respectfully request that the reviewer can agree that our definition of single photons is widely accepted in the literature.

On page 233 of "Quantum Machines: Measurement and Control of Engineered Quantum Systems" (Les Houches Summer School, Oxford University Press, 2011), edited by M. Devoret, B. Huard, R. Schoelkopf, L. Cugliandolo, it is explicitly pointed out by S. Girvin that:

"One often hears the confusing statement that the noise added by an amplifier is a certain number N of photons. This means that the excess output noise produces a flux of N photons per second in a 1 Hz bandwidth."

"It (N) represents the number of photons passing a given point per unit time per unit bandwidth."

The referee is also concerned that our cryogenic detector cannot capture the individual single-photon emission events, which is the energy leakage of a single-excitation standing wave. We feel confused by this concern because we do not measure or claim to measure each individual single-photon emission events. Note that we have written a whole paragraph to clarify this point (lines 134–141). In this study, we measure the second-order correlation function of a propagating field which contains a large amount of photon emission events.

That being said, maybe we still need to clarify why a cryogenic detector is useful if it cannot resolve individual single-photon emission events. Generally speaking, the nanobolometer, including ours [Nature 586, 47-51 (2020)] and other designs such as TES [APL 73, 735-737 (1998)] and KID [Nature 425, 817-821 (2003)], is the most sensitive power detector in the microwave regime. These cryogenic detectors have been widely applied in many research fields, such as the cosmic microwave background (CMB) radiation measurements [Nature 404, 955-959 (2000)]. Because of its high sensitivity, integrability, and low noise, it has

great potential in changing the paradigm of microwave measurements in superconducting quantum circuits that is currently based on cryogenic amplifiers and room-temperature detectors [Nat. Electron. 7, 288-298 (2024)]. The correlation measurement shown in this study pushes bolometry experiments for the first time into the quantum regime, and opens up multitudes of possibilities already in their existing use cases. For example, in astrophysics, it may contribute to resolving a possible distortion of the CMB frequency spectrum from a pure blackbody radiator. In superconducting quantum circuits, it may allow the measurement of correlation functions beyond the 2nd order and the tomography of arbitrary photon states without the low photon number truncation.

In the revised manuscript, we have strengthened the potential significance in the Discussion.

(Remarks on code availability):

I was able to install and run the code.

Reply: We thank the reviewer for verifying our code.

NCOMMS-24-47168-B – Reply to Reviewer #3

Reviewer: In the rebuttal, the authors compare their sensitivity to room temperature detectors which inherently have more dark noise. However, if we compare their device to other cryogenic detectors, we find several examples with better sensitivity [1-5]. Note that in these references, better sensitivity is demonstrated both for radiation coming from mK-scale black body sources, and from other coherent light sources properly normalized to photons/(s × Hz).

Reply: We respond that Refs.1–5 have not used cryogenic detectors for correlation measurement, instead they amplify cryogenic signals and detect them at room temperature. Therefore, the corresponding criticisms do not apply to our study which, to the best of our knowledge, is the first demonstration of correlation measurement of propagating microwave photons with a cryogenic detector.

References 1–5 are publications from the research group where one of the authors, Qi-Ming Chen, has carried out his doctoral training. Indeed, Qi-Ming Chen is also the co-author of Refs.1 and 4. Thus Dr. Chen is well aware of the details of these works, which emphasizes our statement above. In the experiments of Refs.1–5, the signal is amplified at multiple stages and recorded by ADCs at room temperature. Specifically, they are Acqiris-DC440, Acqiris-DC440, Innovative Integration-X6-250m, NI-5782, NI-5782 from Refs.1–5. The major difference is that the photon correlation function in Refs.1–5 can only be inferred from room-temperature measurements with noise-removal data processing techniques, while in our case the correlation is captured at millikelvin by the cryogenic detector and we simply read the position and linewidth of the resonance of the cryogenic detector.

Reviewer: Additionally, the authors claim that the bandwidth is defined by the minimum bandwidth over which power is absorbed. It is true that the nanobolometer absorbs power over a larger bandwidth and could potentially have a larger measurement bandwidth. However, in the measurement presented by the authors, the nanobolometer response is sent through a series of amplifiers and recorded by a room temperature detector where additional filtering is utilized. This additional filtering will limit the response of the detector to a new bandwidth dependent on the overall measurement setup. An analysis of the autocorrelations from a standard source could help the authors determine this bandwidth (see Planck spectroscopy [2]). The key point is that this bandwidth is dependent on the features of the setup which measures the nanobolometer response, not on the bandwidth over which power is absorbed.

Reply: We respectfully disagree that the bandwidth of the input field depends on the bandwidth of the probe field that is used to read out the measurement results of the cryogenic detector. The two fields are indeed independent from each other with also vastly different frequencies.

Again, we emphasize that the photon detection process is implemented by the cryogenic detector itself, i.e., the nanobolometer. The room-temperature setup is used to probe the resonance shift and linewidth of the cryogenic detector that provides the information of input photon correlation. The linewidth is not a time-dependent variable and does

not depend on how an experimentalist measures it. Moreover, the probe field we use to measure the linewidth is not electrically coupled to the input field. This technique is fundamentally different from the conventional method where the detection is implemented at room temperature and the photon correlation is obtained from the amplified input field, which is the key point of novelty of our work.

Reviewer: The authors further argue that the aspects of their measurement setup are not relevant to photon correlation measurements. Specifically, they argue that the linewidth is not a time-dependent variable. However, their data in Fig. 2a directly contradicts this, as noise in the resonance frequency is clearly evident below the fitting. Additionally, correlation measurements are generally considered to be dependent on the measurement bandwidth (see [6] eq. 14.8-22). This is why other techniques [3,4] have characterized their measurement bandwidth using autocorrelation techniques such as Planck spectroscopy, and why the initial paper on microwave Planck spectroscopy cites [6].

Reply: Firstly, Fig. 2a shows the minimum point of the reflection response but does not exhibit the linewidth, as denoted by the y-axis. Secondly, each data point of Fig. 2a is one independent measurement, and they are not time correlated. Thirdly, Fig. 2a is measured at different power of the probe field, as denoted by the x-axis. In correlation measurement, we fix the probe power to one specific value. We can hardly understand how the reviewer got the conclusion that the linewidth is time-dependent according to Fig. 2a.

Of course all experimental data have noise. When we measure the linewidth we take a long-time boxcar average at each probe frequency and we sweep the probe frequency to get the full frequency spectrum. The linewidth is extracted from the measured full spectrum instead of a single-frequency data. The spectrum itself depends on the minimum bandwidth between the source and the cryogenic detector. It cannot depend on the linewidth of the weak probe field that we use to measure it.

Reviewer: Lastly, the authors mention the units of photons/(s \times Hz). I do not argue with the usage of this unit of photons/(s \times Hz) in general. However, calling these quantities 'single photons' is misleading. Photon correlations can expand beyond a single Hz depending on the measurement setup and integration time. Since the measurement modes may be larger than a single Hz, the modes may contain many photons while still having normalized photon levels on the order of 1 photons/(s \times Hz). For example, there may be 31250 photons/s in a 31.25 kHz measurement bandwidth with 4256 modes (i.e. 32 μ s integration time with 133 MHz source bandwidth), this would lead to 1 photons/(s \times Hz), however there would be 7 photons in each mode.

Reply: We first thank the reviewer for agreeing with us that the photons/(s \times Hz) is a proper unit for characterizing propagating photons in our case. Of course, we do not wish to give misleading comments about singleness of photons and have checked the manuscript carefully. We could find only one place where we mention single photons in this context, i.e., the title of the Supplementary Note 1, and have revised it accordingly. This resolves the issue pointed out by the reviewer.

Reviewer: Due to the uncorrected normalization of input light and the lack of demon-

strated sensitivity I do not recommend this manuscript for publication in Nature Communications.

Reply: We of course appreciate the effort the reviewer made in this peer review. However, since we have successfully answered above to the criticisms, we consider that the reasons for this negative recommendations have been eliminated, and hence the recommendation may be changed to positive.

Reviewer:

- [1] Renger, M. et al. Beyond the standard quantum limit for parametric amplification of broadband signals. *npj Quantum Inf.* 7, 1-7 (2021).
- [2] Mariantoni, M. et al. Planck Spectroscopy and Quantum Noise of Microwave Beam Splitters. *Phys. Rev. Lett.* 105, 133601 (2010).
- [3] Goetz, J. et al. Photon Statistics of Propagating Thermal Microwaves. *Phys. Rev. Lett.* 118, 103602 (2017).
- [4] Fedorov, K. G. et al. Experimental quantum teleportation of propagating microwaves. *Sci. Adv.* 7, eabk0891 (2021).
- [5] Gandorfer, S. et al. Two-dimensional Planck spectroscopy for microwave photon calibration. [arXiv:2308.02389 \[quant-ph\]](https://arxiv.org/abs/2308.02389).
- [6] Mandel, L. & Wolf, E. *Optical Coherence and Quantum Optics.* (Cambridge University Press, Cambridge; New York, 1995).

(Remarks on code availability):

The code seems to run.

Reply: We thank the reviewer again for verifying our code.

NCOMMS-24-47168-C – Reply to Reviewer #4

The authors have responded to many of the questions I have had about the work and about its importance. After considering the authors' response and modifications, I recommend this work for publication after some minor revisions.

Reply: We are delighted that we have convinced the reviewer on the solidarity and importance of this study. We thank the reviewer for the recommendation of publication.

After further consideration, I now agree with the authors that the presented technique is fundamentally different from low temperature parametric amplification techniques. As the authors mentioned, the frequency of detection and measurement are completely different. To help readers understand the advancement, the authors should better explain in the manuscript how the work differs from these prior techniques. For example, they could highlight the difference in probe and detection frequencies.

Reply: We added a few clarifying words to the section introducing the way we measure the correlation function and added the following sentence to strengthen this point:

Lines 71-72: “Note that the incident and probe fields are not directly electrically coupled to each other, owing to the ground port between the absorber and the thermometer, and have several gigahertz difference in frequency.”

As the authors mention in their reply, the photon correlation is a key point of novelty in this work. I still believe some additional analysis needs to be done to demonstrate that the probe and detection processes are actually independent. From the presented experiments I find it difficult to tell if this is actually the case. I understand that there is no electrical coupling between the probe field and the input field, however in single photon detection experiments with 1550 nm light, the single photon detectors can be described in a similar manner. In that case, the detector such as a superconducting nanowire single photon detector acts as a bolometer, and produces a GHz signal from incident 1550 nm light.

Additionally, in many photon correlation experiments the readout and detection processes are not independent, and this reflects in the g_2 correlations. In experiments with 1550 nm photons it has been demonstrated that by increasing the measurement time the measured distribution from a thermal source becomes more Poissonian (see [1] Fig. 3f) and therefore g_2 approaches 1. In the manuscript, deviations from theoretically predicted g_2 is observed in Fig. 4, especially in cases with high levels of thermal photons. These deviations do not necessarily demonstrate the probe and detection processes are dependent, but I believe further investigation is necessary. To test this, the authors could verify that when decreasing the FIR filter bandwidth, and when increasing the integration time, the g_2 correlations from a thermal source do not significantly change.

If these g_2 correlations do indeed depend on the FIR filter bandwidth or integration time, then the deviation from theory can be accounted for by treating the photon number as a negative binomial distribution [1-2]. For a negative binomial distribution, the photon number variance will go by $\langle n \rangle + \langle n \rangle^2 / M$ where M is the number of modes with $M = \tau \times \text{BW}$

[3], where tau is the measurement time and BW is the absorbed BW of thermal light. By fitting the measured g^2 values to this distribution, they can find M and therefore τ . Then, $1/\tau$ can replace the FWHM mentioned in Supplementary section 1, and the input light can be properly normalized with this updated value.

Reply: Taking the GHz signal of the superconducting nanowire single photon detector as an example, one may down convert this signal at its central frequency, f_0 , and perform sampling in time domain to reconstruct the signal spectrum (assuming the spectrum is symmetric). The detection bandwidth, which is determined by the FIR filter bandwidth or the integration time, should be larger than the single-side bandwidth of the signal to reconstruct the spectrum.

The key difference in our experiment is that, instead of using a single f_0 and rely on Fourier transform to reconstruct the spectrum, we sweep the down-conversion frequency, f_n with $n = 0, 1, \dots, N$, and look for only the zero-frequency component of the down converted signal. In this case, the detection bandwidth will not affect the reconstructed spectrum because we are directly measuring the Fourier component at each f_n . Because $g^{(2)}(0)$ is obtained solely from the spectrum, the result will also not depend on the detection bandwidth.

Figure 1: Comparison of the measured spectrum with $32 \mu s$ integration (curves) and $16 ns$ integration (dots). The error bars show the standard deviation of the data. The blue to red colors distinguish different coherent input powers.

Because the signal of interest is narrowband while the noise is broadband, we choose a 500 kHz low-pass FIR filter and integrate the result by $32 \mu s$ to improve the signal to noise ratio (SNR). To further convince the reviewer that the detection bandwidth only affects the SNR but not the measured spectrum, here we provide a comparison of two spectra with different integration times. The detection bandwidths differ by a factor of 2048, but the spectrums are identical to each other except for different noise levels as shown by the error bars. This comparison should provide a firm answer that the detection bandwidth affects only the SNR but not the measurement result of $g^{(2)}(0)$. We hope that thanks to this confirmation, the manuscript can be published without further revisions.

- [1] Cheng, R. et al. A 100-pixel photon-number-resolving detector unveiling photon statistics. *Nat. Photon.* 17, 112-119 (2023).
- [2] Katamadze, K. G. et al. Multimode thermal states with multiphoton subtraction: Study of the photon-number distribution in the selected subsystem. *Phys. Rev. A* 101, 013811 (2020).
- [3] Mandel, L. & Wolf, E. *Optical Coherence and Quantum Optics.* (Cambridge University Press, Cambridge; New York, 1995).

(Remarks on code availability):

The code seems to run.

Reply: Again, we thank the reviewer for verifying our code.